# UnCLe SAM: Unleashing SAM's Potential for Continual Prostate MRI Segmentation

**Amin Ranem**[1]                                        AMIN.RANEM@GRIS.INFORMATIK.TU-DARMSTADT.DE
**Aflam Ahlal**[1]                                        AFHAM.MOHAMED@STUD.TU-DARMSTADT.DE
**Moritz Fuchs**[1]                                        MORITZ.FUCHS@GRIS.INFORMATIK.TU-DARMSTADT.DE
**Anirban Mukhopadhyay**[1]  ANIRBAN.MUKHOPADHYAY@GRIS.INFORMATIK.TU-DARMSTADT.DE
[1] *Technical University of Darmstadt, Karolinenpl. 5, 64289 Darmstadt, Germany*

**Editors:** Accepted for publication at MIDL 2024

## Abstract

Continual medical image segmentation primarily explores the utilization of U-Net and its derivatives within the realm of medical imaging, posing significant challenges in meeting the demands of shifting domains over time. Foundation models serve as robust knowledge repositories, offering unique advantages such as general applicability, knowledge transferability, and continuous improvements. By leveraging pre-existing domain insights, adaptability, generalization, and performance across diverse tasks can be enhanced. In this work, we show how to deploy Segment Anything Model's (SAM) natural image pretraining for the continual medical image segmentation, where data is sparse. We introduce **UnCLe SAM**, a novel approach that uses the knowledge of the pre-trained SAM foundation model to make it suitable for continual segmentation in dynamic environments. We demonstrate that UnCLe SAM is a robust alternative to U-Net-based approaches and showcase its state-of-the-art (SOTA) continual medical segmentation capabilities. The primary objective of UnCLe SAM is to strike a delicate balance between model rigidity and plasticity, effectively addressing prevalent pitfalls within CL methodologies. We assess UnCLe SAM through a series of prostate segmentation tasks, applying a set of different CL methods. Comparative evaluations against the Lifelong nnU-Net framework reveal the potential application of UnCLe SAM in dynamically changing environments like healthcare. Our code base is available at `https://github.com/MECLabTUDA/UnCLeSAM/`.

**Keywords:** Continual learning, Foundation Model, Segment Anything Model

## 1. Introduction

Continual learning (CL) holds immense significance in *safety-critical applications* of Deep Learning. This is evident in healthcare, where models must adapt to data changes over time while maintaining high performance on older data (Gonzalez et al., 2020). Traditional *U-Net architectures encounter difficulties* in seamlessly adapting to domain changes in data distribution, particularly when faced with new imaging protocols or variations in patient populations or diseases (Sanner et al., 2021; Derakhshani et al., 2022; González et al., 2022; Fuchs et al., 2022). The challenge lies in training models that exhibit superior performance when using datasets with limited temporal availability. Finding a good trade-off between rigidity which hinders learning new tasks and plasticity causing catastrophic forgetting on previous tasks is therefore important (Kirkpatrick et al., 2017; Hadsell et al., 2020;

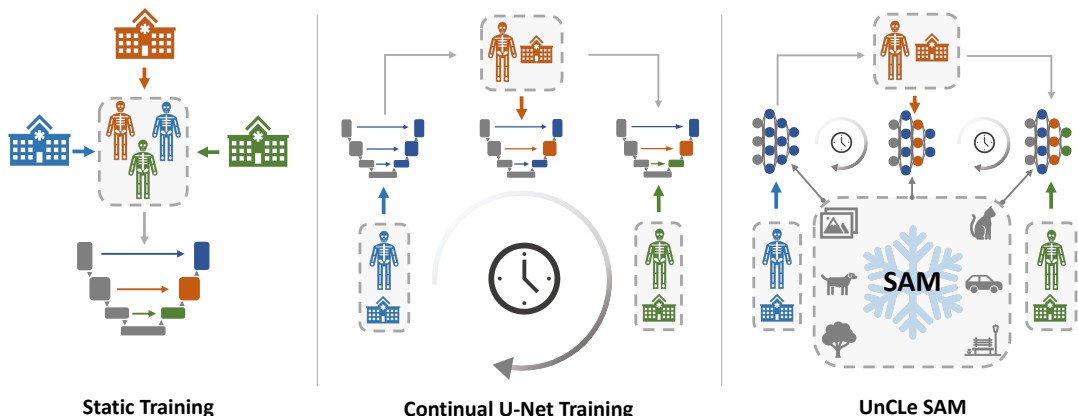

Figure 1: Unlike traditional static training (left), continual U-Net training (middle) involves time-limited access to training data. Data arrives sequentially, and the model lacks access to previous data. In contrast, UnCLe SAM (right) continuously adapts the adapter with sequentially arriving data while benefiting from SAM's pre-trained knowledge base.

De Lange et al., 2021). Existing CL methods, when applied to medical data, often result in segmentations that fall short of basic semantic standards like semantic coherence over time (Ranem et al., 2022; González et al., 2023).

We introduce **UnCLe SAM** (Unleashing Continual Learning for SAM), a novel approach that leverages the Segment Anything Model (SAM) (Kirillov et al., 2023) for *enhanced domain adaptation* in continuous medical setups. Our method involves continually adapting the prompt for SAM while leveraging the knowledge base of SAM without a full re-training as in MedSAM (Ma and Wang, 2023).

The Lifelong nnU-Net framework and other CL methods, such as Elastic Weight Consolidation (EWC) (Kirkpatrick et al., 2017), Riemannian Walk (RWalk) (Chaudhry et al., 2018), or basic replay methods like iCARL (Rebuffi et al., 2017), struggle to adequately adapt to changing domains (González et al., 2023). Using replay, regularization, or knowledge distillation has its advantages and disadvantages when it comes to domain shifts in continuous setups. For instance, there is a rigidity/plasticity trade-off or computational burden on performance. Exploring CL by using foundation models, until now, remained an unexplored approach, Figure 1.

*UnCLe SAM strategically addresses challenges for domain adaptation* known when applying U-Net-based architectures. U-Nets encounter challenges in maintaining segmentation accuracy amidst variations in imaging protocols or discrepancies in patient populations, resulting in compromised performance and reduced reliability of the model in dynamic clinical scenarios (Gonzalez et al., 2020; Ranem et al., 2022; Sanner et al., 2021). Leveraging the robust knowledge from SAM while continually adapting the prompting adapter reduces such challenges.

In this work, we use the pre-trained SAM architecture for continual prostate MRI segmentation that *leverage the foundation model's knowledge to properly adapt to shifting domains.* Rather than attempting to apply regularization to the network, we opt to freeze

certain architectural components such as the Vision Transformer, i.e., SAM's Encoder while *continually adapting the prompt for SAM*. With this approach, the learned visual representation can be transfered to different domains (Ma and Wang, 2023).

SAM's ability to segment anything in diverse contexts becomes a valuable asset for continual adaptation, ensuring that the model maintains high segmentation performance across evolving datasets with simple fine-tuning techniques. By using a pre-trained ResNet-50 network, (He et al., 2016) as an Adapter to continually updating the SAM prompt, UnCLe SAM effectively handles challenges that come with domain adaptations commonly faced by U-Net-based architectures. *UnCLe SAM does not require a long time to train* on a new domain, which makes it superior in terms of applicability while achieving SOTA performance. To validate our approach, we focus on the critical task of prostate segmentation for T2-weighted MRIs, which plays an important role in prostate cancer diagnosis and treatment planning. Our contributions are three-fold: We (1) introduce a **Continual prompting of foundation model** for medical image segmentation, that can (2) successfully respond to **domain adaptation** by achieving (3) **superior performance than** Lifelong nnU-Net Framework.

## 2. Methodology

**Fundamentals**   We start by introducing some key terminology: $\Omega \subset \mathbb{R}^3$ defines a 3D spatial domain as we work with three-dimensional Magnetic Resonance (MR) scans. $\mathcal{T}_i \subset \Omega_{\mathcal{T}}$ is referred to a single task $i$, whereas $\Omega_{\mathcal{T}}$ represents a set of tasks. A stage $j$ in a continual setup defines the process of training the model on task $\mathcal{T}_j$ after it has been trained on all previous $\{\mathcal{T}_1, \ldots, \mathcal{T}_{j-1}\}$ tasks using some CL method.

**Basic components of SAM**   SAM (Kirillov et al., 2023), consists of a Vision Transformer (ViT) (Dosovitskiy et al., 2020) as its core feature extractor and a segmentation head in form of a mask decoder, responsible for generating precise segmentation masks. Trained on a large database of general images, SAM has garnered a robust knowledge base that facilitates its adaptability across various domains opening doors for continual setups.

The ViT feature extractor within SAM effectively captures visual information from input images, creating detailed embeddings. SAM's segmentation head complements the ViT feature extractor by processing the embeddings to produce detailed segmentation masks. This mask decoder is trained to accurately delineate regions of interest within the input images. Moreover, the segmentation head can makes use of different prompts such as 2D points and bounding boxes to further guide the segmentation process. SAM was trained on a large database of general images to establish a strong foundation for adaptable segmentation tasks. This foundation enables SAM to excel in various domains, making it particularly well-suited for continual setups where adaptability is paramount.

**UnCLe SAM: Continual Prompting for Enhanced Adaptability**   UnCLe SAM builds upon the foundation of SAM by introducing a novel approach to enhance adaptability over time. Since SAM generalizes to different domains, UnCLe SAM enhances this adaptability by introducing continual prompting using a ResNet-50 adapter. The architecture of UnCLe SAM is carefully crafted to leverage the global knowledge stored within the pre-trained SAM, offering adaptability in dynamic environments like healthcare. A key

aspect of our methodology is the decision to keep SAM's ViT backbone frozen, ensuring consistent feature extraction across different datasets and imaging modalities. This approach not only enhances feature extraction reliability but also lays the groundwork for seamless adaptation to changing domain characteristics, see Figure 2. By leveraging the frozen ViT backbone, we implement a pre-processing step to extract embeddings from both training and testing sets, aligning with the approach proposed by MedSAM (Ma and Wang, 2023).

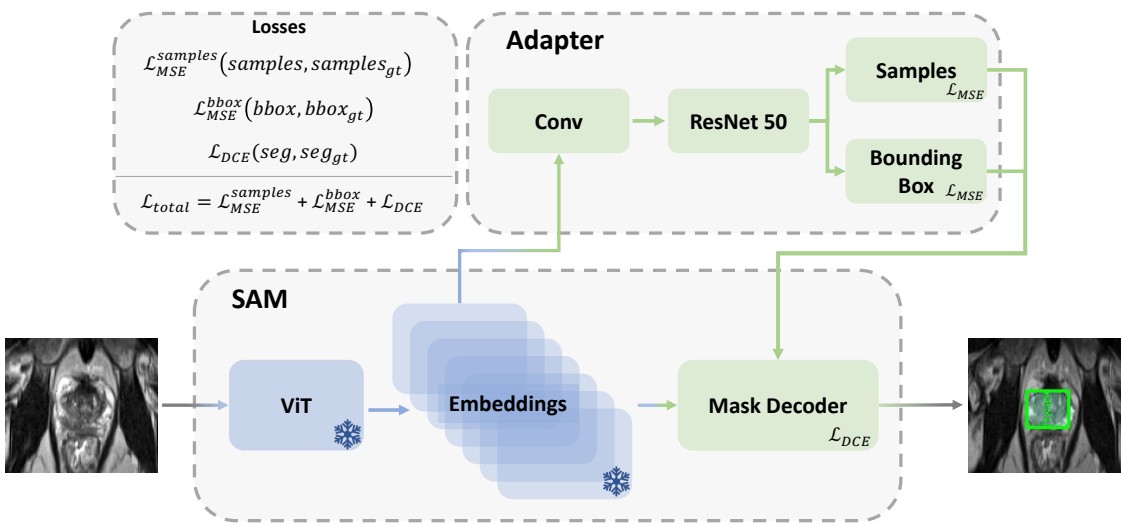

Figure 2: UnCLe SAM model for medical image segmentation using a ResNet-50 as an Adapter to build the input prompt for the base SAM backbone.

A key component of UnCLe SAM lies in the continuous adaptation of the ResNet-50 adapter, which plays a crucial role in guiding the segmentation process by generating adaptive prompts. These adaptive prompts are based on the embeddings extracted by the ViT feature extractor. By incorporating a transposed convolutional layer, the ResNet-50 adapter effectively translates the embeddings into actionable prompts, improving segmentation accuracy. Based on the extensive ablations of SAM in radiology (Ranem et al., 2023), the proposed adapter is designed to predict 100 2D points and four coordinates representing a bounding box.

Moreover, UnCLe SAM is designed to adapt to changing domain characteristics over time. While SAM's basic components provide a strong foundation for segmentation tasks, UnCLe SAM's continual prompting mechanism ensures that the model can dynamically adjust to evolving datasets and domain shifts. This adaptability is essential for maintaining high performance across diverse medical imaging environments.

In summary, *UnCLe SAM's architectural composition integrates the robustness of pretrained SAM and the feature extraction capabilities of ResNet-50*, forming a comprehensive base model for domain adaptation in medical image segmentation over time. By strategically freezing SAM's ViT backbone, coupled with the ResNet-50 Adapter, UnCLe SAM is the first method to effectively combine the strengths of both a foundation model and CL to achieve accurate and adaptable segmentation results for medical segmentation tasks.

## 3. Experimental Setup

**Datasets** We explore the problem of continual image segmentation for prostate MRIs. To ensure reproducibility, we use only openly available datasets, whereas every data source acts as one task $\{\mathcal{T}_1, \ldots, \mathcal{T}_n\}$. Table 1 provides a summary of the core characteristics of the data.

| Dataset | UCL | I2CVB | ISBI | DecathProst |
|---|---|---|---|---|
| # Cases | 13 | 19 | 30 | 32 |
| Resolution | [24 384 384] | [64 384 384] | [19 384 384] | [19 316 316] |
| Spacing | [3.3 0.5 0.5] | [1.3 0.5 0.4] | [3.7 0.5 0.5] | [1.0 1.0 1.0] |

Table 1: Image and label characteristics of the used prostate datasets.

The prostate data corpus consists of four publicly available T2-weighted MRI datasets as provided in the Multi-site Dataset for Prostate MRI Segmentation Challenge for sites A (ISBI), C (I2CVB) and D (UCL) and DecathProst from the Medical Segmentation Decathlon (Litjens et al., 2014; Bloch et al., 2015; Lemaître et al., 2015; Liu et al., 2020a,b; Antonelli et al., 2021). For all datasets, we randomly divide 20% of the data for test purposes and maintain this split across all experiments.

**Training setup** All nnU-Net (Isensee et al., 2021) experiments train for 250 epochs with 250 steps each using the Lifelong nnU-Net framework (González et al., 2023) with default optimizer and scheduler. SAM experiments also run for 250 epochs using Adam optimizer with weight decay of $1e^{-4}$, learning rates of $1e^{-4}$ and $1e^{-3}$ for the SAM segmentation head and the Adapter respectively. Our loss function combines Dice-Cross-Entropy (DCE, $\mathcal{L}_{DCE}$) from (Consortium, 2020) and Mean-Squared-Error (MSE, $\mathcal{L}_{MSE}$) with early stopping (patience of 15). The MSE loss-term is used for predicted samples ($\mathcal{L}_{MSE}^{samples}$) and bounding box coordinates ($\mathcal{L}_{MSE}^{BBox}$): $\mathcal{L}_{SAM} = \mathcal{L}_{DCE} + \mathcal{L}_{MSE}^{samples} + \mathcal{L}_{MSE}^{BBox}$. All models are trained on a single NVIDIA A40 GPU (48 GB).

**Metrics** For every CL setup, we report the mean Dice and standard deviation across the test images from all tasks $\{\mathcal{T}_i\}_{i \leq |\Omega_{\mathcal{T}}|}$ as well as average forwards (FWT) and backwards (BWT) transferability (Díaz-Rodríguez et al., 2018). FWT measures the impact of the current training stage $\{\mathcal{T}_i\}_{i \leq |\Omega_{\mathcal{T}}|}$ on test data from an untrained stage $\mathcal{T}_j$ ; $j > i$. BWT, on the other hand, indicates the amount of maintained knowledge on test samples from $\mathcal{T}_j$ during training on different stages $\{\mathcal{T}_i\}_{i \leq |\Omega_{\mathcal{T}}|}$ ; $j < i$ over time. Models that achieve a higher FWT have high plasticity and are able to learn new knowledge, while models with a higher BWT maintain most knowledge from previous tasks, i.e. prevent catastrophic forgetting. More information on the CL metrics can be found in the Appendix B.

**Baselines** To get a proper evaluation of our approach, we compare against conventional sequential training, rehearsal training, and two well-known CL methods: EWC (Kirkpatrick et al., 2017) and RWalk (Chaudhry et al., 2018). For both CL methods we are inspired by the Lifelong nnU-Net (González et al., 2023) hyperparameter setup for all our experiments (EWC: $\lambda = 0.4$, RWalk: $\alpha = 0.9, \lambda = 0.4$).

## 4. Results

### 4.1. Continual learning performance

In this section, we compare UnCLe SAM with sequential Lifelong nnU-Net and two established CL methods – EWC and RWalk. Additionally, we evaluate against the upper bound of rehearsal training, which involves storing randomly 20% from each task. Rehearsal serves as an upper bound for Lifelong nnU-Net but is impractical due to privacy policy constraints on storing patient images.

| Method | Fixed param | Tuned param | Dice ↑ [%] | BWT ↑ [%] | FWT ↑ [%] | # Epochs ↓ | Runtime ↓ [sec] |
|---|---|---|---|---|---|---|---|
| Sequential$_{\text{nnU-Net}}$ | – | – | $49.44 \pm 28.82$ | $-52.96 \pm 15.07$ | $-52.81 \pm 5.05$ | 1000 | 193 |
| Sequential$_{\text{UnCLe SAM}}$ | | | $\mathbf{78.38 \pm 11.67}$ | $\mathbf{-14.27 \pm 8.91}$ | $\mathbf{-21.27 \pm 10.96}$ | **113** | **43** |
| EWC$_{\text{nnU-Net}}$ | – | $\lambda = 0.4$ | $39.34 \pm 32.03$ | $-46.77 \pm 12.16$ | $-52.72 \pm 16.90$ | 1000 | 200 |
| EWC$_{\text{UnCLe SAM}}$ | | | $77.77 \pm 12.16$ | $-16.85 \pm 10.13$ | $-22.40 \pm 10.86$ | 123 | 46 |
| RWalk$_{\text{nnU-Net}}$ | $\alpha = 0.9$ | $\lambda = 0.4$ | $52.48 \pm 26.19$ | $-48.62 \pm 13.42$ | $-48.73 \pm 9.52$ | 1000 | 196 |
| RWalk$_{\text{UnCLe SAM}}$ | | | $77.31 \pm 13.42$ | $-16.08 \pm 12.86$ | $-23.17 \pm 11.41$ | 120 | 48 |
| Rehearsal$_{\text{nnU-Net}}$ | – | – | $60.90 \pm 21.62$ | $-37.45 \pm 11.60$ | $-39.83 \pm 7.92$ | 1000 | 269 |

Table 2: CL performance of the final model; mean Dice, BWT and FWT over all tasks including standard deviation, total amount of trained epochs and average runtime per epoch in seconds; best values are marked in bold.

Table 2 and Figure 3 show that Lifelong nnU-Net achieves certain benefits depending on which CL method is used, however gets significantly outperformed by UnCLe SAM.

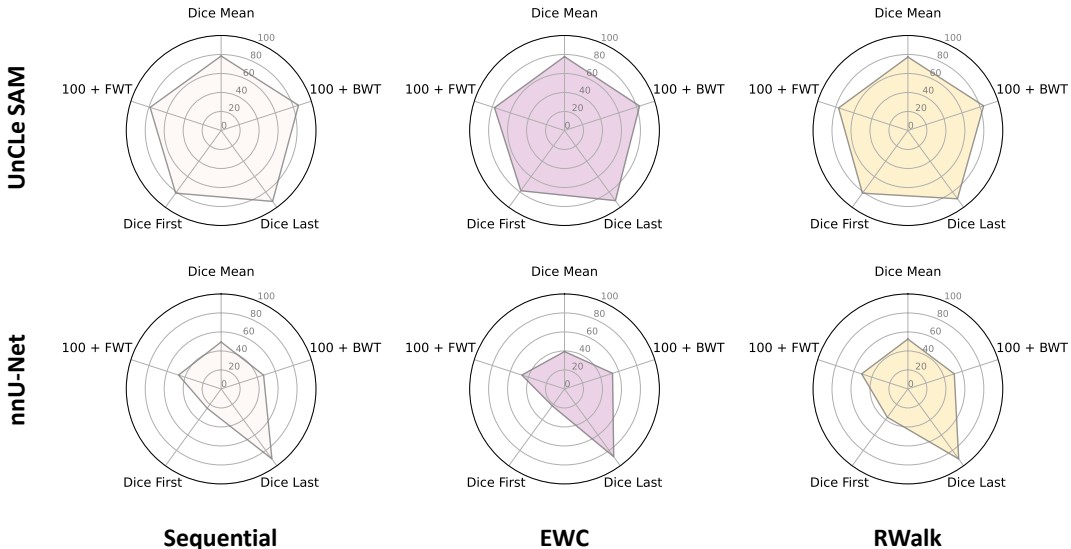

Figure 3: Segmentation performance as Dice using different CL methods for UnCLe SAM and Lifelong nnU-Net; the larger the area the better.

UnCLe SAM demonstrates superiority over Lifelong nnU-Net as it *successfully leverages the rich knowledge base embedded in the foundation model*, enabling robust adaptation to domain shifts within the data. The 23% and 18% performance increase for BWT and FWT,

compared to the rehearsal upper bound, contrasts with traditional methods, which *struggle to handle domain variations effectively* over time.

## 4.2. Qualitative temporal evaluation

To analyze the robustness of our proposed method, we illustrate segmentation masks in Figure 4 for UnCLe SAM and Lifelong nnU-Net using EWC, RWalk and rehearsal.

UnCLe SAM consistently generates coherent segmentation masks throughout all training stages. In contrast, EWC and rehearsal training for Lifelong nnU-Nets result in low-quality segmentations after training on the last stage 4 $\{\mathcal{T}_4\}$. The reduced performance on the sample scan for later stages illustrates the impact of catastrophic forgetting, where the network excessively adapts to the most recent training data, i.e. being too plastic. UnCLe SAM avoids being either overly rigid or plastic, by achieving a proper balance, providing robust predictions that maintain quality across both early and later training stages.

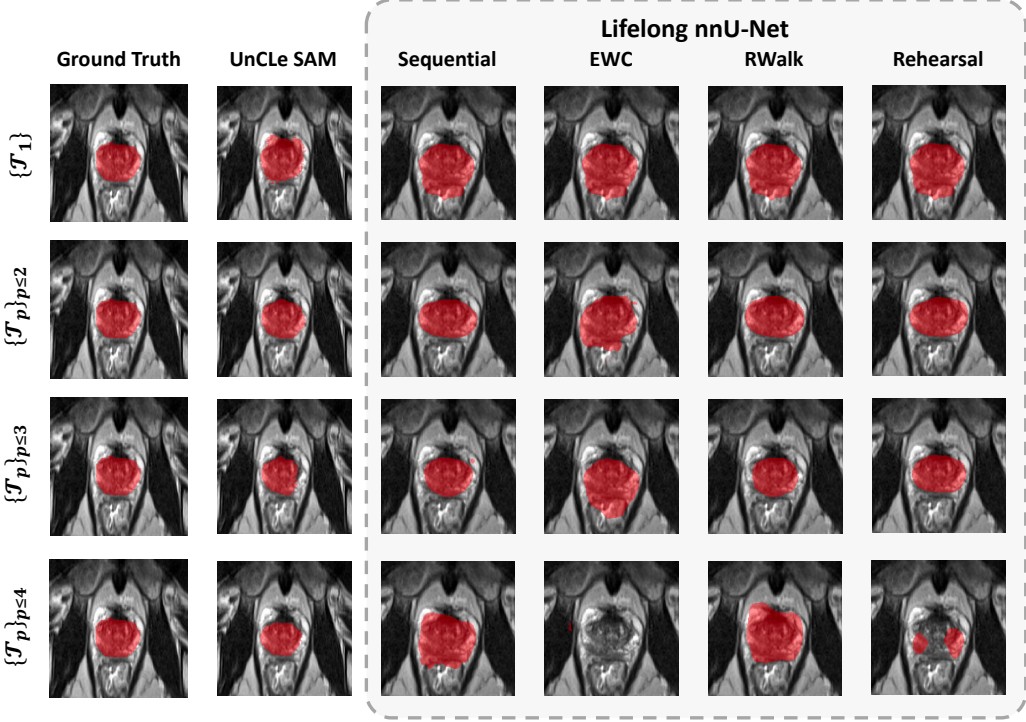

Figure 4: Temporal analysis for sequential, EWC, Rehearsal and UnCLe SAM using Case 14, Slice 14 (37) from $\mathcal{T}_2$.

## 4.3. UnCLe SAM static comparison

In a static training condition, a model is trained on one single task and validated across all existing tasks, as shown in Table 3. Direct comparison between UnCLe SAM and nnU-Net segmentation under static training provide direct insights into a method's generalizability and ability to handle diverse tasks after training on a singular dataset.

| | Trained on | Tested on – Dice $\uparrow \pm \sigma \downarrow$ [%] | | | |
|---|---|---|---|---|---|
| | | UCL | I2CVB | ISBI | DecathProst |
| nnU-Net | UCL | **85.47 ± 6.92** | 23.24 ± 16.8 | 81.47 ± 10.7 | 9.68 ± 11.4 |
| | I2CVB | 57.11 ± 7.57 | **83.06 ± 0.28** | 45.73 ± 20.2 | 1.30 ± 1.53 |
| | ISBI | 81.78 ± 6.15 | 29.06 ± 17.6 | **93.00 ± 1.46** | 52.48 ± 27.5 |
| | DecathProst | 25.24 ± 25.1 | 27.57 ± 1.89 | 59.20 ± 16.3 | **89.25 ± 1.78** |
| UnCLe SAM | UCL | **85.29 ± 3.59** | 51.53 ± 36.1 | 85.57 ± 6.61 | 44.57 ± 21.8 |
| | I2CVB | 85.99 ± 1.83 | **88.11 ± 3.20** | 83.55 ± 8.42 | 80.16 ± 6.52 |
| | ISBI | 84.68 ± 0.93 | 54.84 ± 38.8 | **96.02 ± 1.04** | 82.56 ± 3.79 |
| | DecathProst | 81.55 ± 8.66 | 59.94 ± 36.4 | 79.77 ± 15.1 | **92.27 ± 0.97** |

Table 3: Results for nnU-Net and UnCLe SAM networks trained on every task individually and evaluated across all tasks; Bold values indicate the performance of the baseline on the validation set of the task it has been trained on.

Table 3 demonstrates the *superior performance and greater generalizability of UnCLe SAM compared to nnU-Net* for prostate segmentation. The method consistently achieves higher Dice scores across diverse datasets, achieving proper generalizability, highlighting UnCLe SAM's robustness and adaptability. It showcases promising potential for domain adaptation and continual learning in medical image segmentation, providing a stable and adaptable solution across diverse datasets. For additional results, we refer the reader to Appendix A.

## 5. Conclusion

We propose *UnCLe SAM*, a *novel approach* that leverages the knowledge base of the pre-trained SAM *foundation model* to address domain adaptation challenges in *continual medical image segmentation*. By leveraging SAM's robust capabilities, our method achieves *superior adaptability and performance* compared to traditional U-Net architectures like the Lifelong nnU-Net framework. Through extensive evaluation of a set of four different prostate datasets, UnCLe SAM demonstrates its effectiveness in maintaining knowledge from early stages while adapting to evolving datasets over time. Our approach not only *outperforms existing methods* in terms of segmentation accuracy in a continuous setup but also offers a more generalizable solution, showcasing a significant performance improvement even when trained statically with data from a single site. UnCLe SAM paves the way for a *balanced approach between rigidity and plasticity* in continual learning setups without using actual CL methods like EWC while achieving better results than rehearsal. By releasing our code base, we hope to inspire research in CL that goes beyond traditional U-Net-based segmentation for medical settings by leveraging the knowledge base of foundation models like SAM.

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

## Appendix A. Base model performance

Table 3 from the main manuscript provides the Dice scores with standard deviation for every trained baseline evaluated across all tasks. Figure 5 visualizes them in form of confusion matrices.

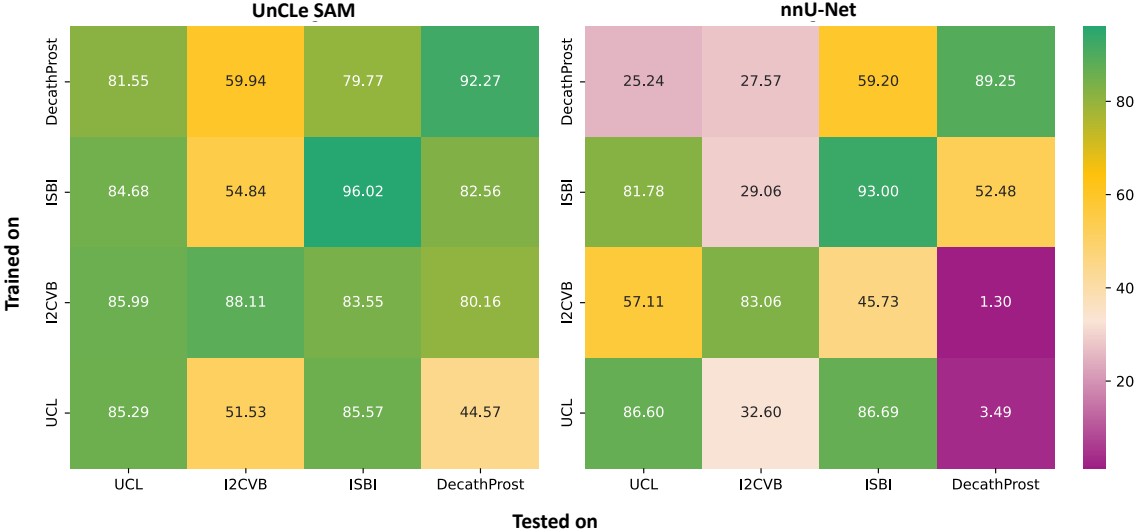

Figure 5: Confusion matrices based on Dice score for UnCLe SAM (left) and nnU-Net (right) across different datasets.

## Appendix B. Continual learning metrics

In this work, BWT and FWT are defined as follows (Díaz-Rodríguez et al., 2018). Let $\mathcal{T}_i$ be a specific task:

FWT is defined as

$$\text{FWT}\left(\mathcal{T}_i\right) = \text{Dice}\left(\mathcal{M}_{[\mathcal{T}_1,\ldots,\mathcal{T}_{i-1}]}, \mathcal{T}_i\right) - \text{Dice}\left(\mathcal{M}_{[\mathcal{T}_i]}, \mathcal{T}_i\right), \tag{1}$$

where $\mathcal{M}_{[\mathcal{T}_1,\ldots,\mathcal{T}_i]}$ is a network trained on stages $\{1,\ldots,p\} \leq |\Omega_{\mathcal{T}}|$ and $\text{Dice}(\mathcal{M}_{[\mathcal{T}_1,\ldots,\mathcal{T}_j]}, \mathcal{T}_i)$ indicates the Sørensen–Dice coefficient from a network trained on stages $\{1,\ldots,j\}$ evaluated on dataset $p$.

BWT is defined as

$$\text{BWT}\left(\mathcal{T}_i\right) = \text{Dice}\left(\mathcal{M}_{[\mathcal{T}_1,\ldots,\mathcal{T}_i,\ldots,\mathcal{T}_n]}, \mathcal{T}_i\right) - \text{Dice}\left(\mathcal{M}_{[\mathcal{T}_1,\ldots,\mathcal{T}_i]}, \mathcal{T}_i\right), \tag{2}$$

FWT for the last model state as well as BWT for the first model state is not defined.

## Appendix C. Continual learning performance

Table 2 from the main manuscript provides the CL performance over all used methods using mean Dice, BWT and FWT. Table 4 provides the actual Dice scores and standard deviation of each method across all four tasks after the network was trained on all stages in a continous manner.

| Method | Tested on – Dice $\uparrow \pm \sigma \downarrow$ [%] | | | |
|---|---|---|---|---|
| | UCL | I2CVB | ISBI | DecathProst |
| Sequential$_{nnU\text{-}Net}$ | $25.06 \pm 5.61$ | $20.22 \pm 3.95$ | $61.40 \pm 15.7$ | $91.06 \pm 1.61$ |
| Sequential$_{UnCLe\ SAM}$ | $81.55 \pm 8.66$ | $\mathbf{59.94 \pm 36.4}$ | $79.77 \pm 15.1$ | $\mathbf{92.27 \pm 0.97}$ |
| EWC$_{nnU\text{-}Net}$ | $21.98 \pm 3.92$ | $2.17 \pm 2.57$ | $45.01 \pm 24.9$ | $88.20 \pm 1.30$ |
| EWC$_{UnCLe\ SAM}$ | $78.36 \pm 8.57$ | $58.26 \pm 36.5$ | $83.19 \pm 10.7$ | $91.25 \pm 1.01$ |
| RWalk$_{nnU\text{-}Net}$ | $36.86 \pm 7.82$ | $21.67 \pm 4.56$ | $60.33 \pm 18.9$ | $91.07 \pm 1.54$ |
| RWalk$_{UnCLe\ SAM}$ | $\mathbf{81.56 \pm 4.95}$ | $54.51 \pm 38.5$ | $\mathbf{84.34 \pm 5.47}$ | $88.85 \pm 3.20$ |
| Rehearsal$_{nnU\text{-}Net}$ | $35.39 \pm 30.7$ | $46.59 \pm 3.03$ | $70.30 \pm 16.00$ | $91.33 \pm 1.45$ |

Table 4: Results for nnU-Net and UnCLe SAM final networks trained all tasks sequentially, with EWC and RWalk, evaluated across all tasks; Bold values indicate the best performance.

## Appendix D. Workflow comparison of models

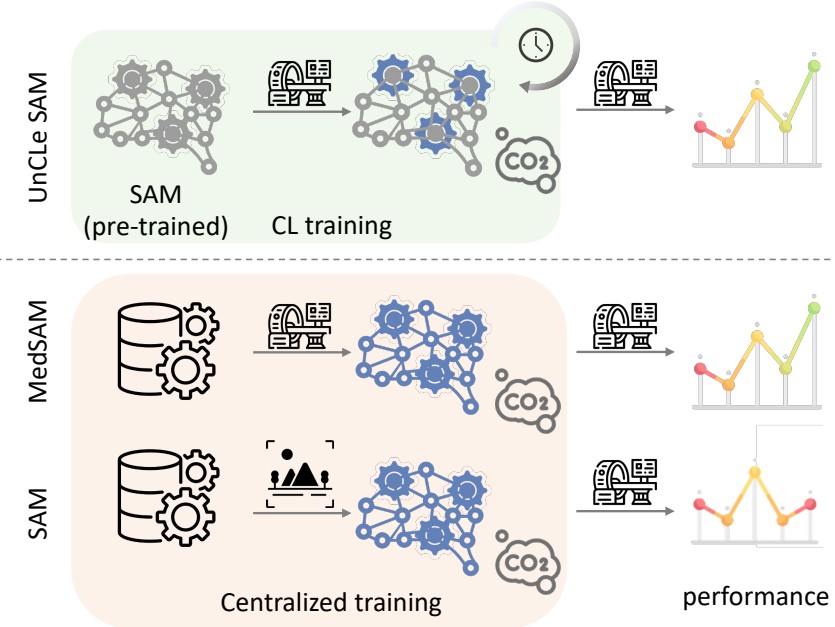

Figure 6: Comparison of workflow methodologies for SAM, MedSAM, and UnCLe SAM. SAM and MedSAM adopt a centralized training approach, whereas SAM fails to perform good in medical use cases. In contrast, UnCLe SAM utilizes a continuous training paradigm, facilitating CL.

Figure 6 illustrates the workflow process for SAM, MedSAM and UnCLe SAM, highlighting their distinct training methodologies. SAM and MedSAM are trained centralized from scratch, while having an increased CO2 emission. Additionally, having centralized training

of MedSAM may inadvertently contain samples from publicly used datasets, resulting in data leakage and privacy violations in CL comparisons. In contrast, UnCLe SAM utilizes a continuous training paradigm, minimizing $CO_2$ emissions by leveraging the pre-trained SAM model and enabling adaptation to evolving data leveraging the pre-trained SAM.

