# OpenReview forum: "UnCLe SAM: Unleashing SAM’s Potential for Continual Prostate MRI Segmentation"
_MIDL.io/2024/Conference — MIDL 2024 Poster_

### Official Review · Reviewer_pvfF · 2024-02-28

**Confidence:** 5
**Preliminary Rating:** 4
**Recommendation:** Oral
**Final Rating:** 5

**Summary:**

The authors present a CL method for MRI segmentation based on the pre-trained Segment Anything Model with an appended ResNet50 adapter that improves domain adaptation and achieves superior performance than the "state-of-the-art" Lifelong nnUNet.

**Strengths:**

The paper is well-written and gives a good overview of common training approaches, and the problem they tackle. They give a detailed description of their proposed model, which consistently outperformes nnUNet and is faster to train. The static comparisons between UnCLe SAM and the nnUNet easily highlight the merit of the proposed model.

**Weaknesses:**

The authors point out, that their model outperforms the 'SOTA' Lifelong nnUNet, however their referenced publication (González et. al.) does not claim state-of-the-art results. The authors should reference what they base their assumption on that the Lifelong nnUNet is state-of-the-art, or preferrably, rephrase and simply claim that they largely outperform the nnUNet model, which in itself is a great achievement.
Most of the paper is well-written and very detailed, but there are parts where the text is too high level, and the motivation behind some paragraphs is unclear. I will give more details later.

**Detailed Comments:**

The author names are witheld in the submitted manuscript, however they are visible in the submission. I'm only commenting on this, so the authors can make sure that they are using the most recent latex templates.
I'm very much looking forward to the publicly available code repository.

**Justification Of Final Rating:**

Thank you for addressing my concerns, and taking an active part in the rebuttal. Although I agree with some of the other reviewers about the added novelty of the proposed method, I believe that showing the clear advantages of using pre-trained models is an important one, especially in scenarios where the model was pre-trained on images from another domain. Highlighting these advantages can help researchers develop models in areas where data is scarce, and the use of a publicly available model helps with the reproducibility as well.

**Justification Of The Preliminary Rating:**

The problem the authors tackle is important, their proposed method performs very well, and is fast to train. Their evaluations are thorough, and I believe a few adjustments to their manuscript could better highlight the quality of their work.

**Questions To Address In The Rebuttal:**

- Do the authors have an assumption on how the pre-trained SAM model in itself (without the adapter) would perform for MRI segmentation? If so, it would be interesting to include the results in the static evaluations. A strong motivation between the much better performance of UnCLe SAM over the static nnUNet could be simply that SAM has been extensively pre-trained before training here.
- The need for domain adaptation is clear, and it's a strong motivation of the paper, however the lack of access to previous training data is less obvious. Could the authors explain a scenario where access to previous training data is lost?
- The paragraph "Basic components of SAM" is very high-level and the length of the paragraph isn't really motivated. I believe shortening it would improve the focus of the manuscript.

**Special Issue:**

No

---

> ### Author Response · Authors · 2024-03-14
>
> Thank you for your insightful comments, which have helped us improve the clarity and focus of our manuscript. If you have any further suggestions or questions, please let us know. We addressed each of your points:
>
> 1. **Performance of Pre-trained SAM Model**: In our recent work on "Exploring SAM Ablations for Enhancing Medical Segmentation in Radiology and Pathology," we observed that even when SAM is pre-trained and used with ground truth labels, it fails to achieve very high performances for radiology cases. For example, the highest performance achieved in the BraTS dataset was 84%. This insight suggests that the strong performance of UnCLe SAM over the static nnU-Net cannot solely be attributed to extensive pre-training of SAM.
>
> 2. **Explanation of the Lack of Access to Previous Training Data**: Consider the limited availability of patient data due to factors such as time constraints or changes in population or acquisition methods. In the case of prostate segmentation, changes in acquisition methods, such as the use of prostate coils that were previously used but are now outdated, can contribute to severe domain shifts. Additionally, privacy constraints over time may prohibit the long-term storage of patient scans, further limiting access to previous training data.
>
> 3. **Simplification of the "Basic Components of SAM" Paragraph**: We have reformulated and shortened the paragraph to make it easier to understand and to improve the focus of the manuscript.

---

### Official Review · Reviewer_TAJc · 2024-02-28

**Confidence:** 4
**Preliminary Rating:** 5
**Recommendation:** Oral

**Summary:**

This paper introduces UnCLe SAM, a novel approach that leverages the Segment Anything Model (SAM) for continual prostate MRI segmentation. UnCLe SAM strategically addresses challenges faced by traditional U-Net-based architectures in maintaining segmentation accuracy amidst variations in imaging protocols and patient populations, presenting a promising approach for continual medical image segmentation in dynamic clinical scenarios.

**Strengths:**

The idea to leverage SAM for continual segmentation is novel. The paper is well-written and conducted comprehensive analysis and comparison to demonstrate the effectiveness. The demonstrated superior performance compared to the state-of-the-art methods adds significant value to the proposed approach.

**Weaknesses:**

Overall the method and experiments are sufficient. I have several minor suggestions.
- It would be better to compare with original SAM and other medical variants like MedSAM.
- What does 'UnCLe' means in the name? U-Net-based Continual Learning? It would be better to specify this in the revised version.

**Detailed Comments:**

See above

**Justification Of The Preliminary Rating:**

The idea to leverage SAM for continual segmentation is novel. The paper is well-written and conducted comprehensive analysis and comparison to demonstrate the effectiveness. The demonstrated superior performance compared to the state-of-the-art methods adds significant value to the proposed approach.

**Questions To Address In The Rebuttal:**

See above.

**Special Issue:**

Yes

---

> ### Author Response · Authors · 2024-03-14
>
> Thank you for your insightful comments. We have considered your suggestions and addressed them accordingly in the revised manuscript. Regarding our used term “Segmentation Head”, this only includes the standalone mask decoder, not the prompt encoder.
>
> 1. **Comparison with Original SAM and Other Medical Variants**: We acknowledge the importance of comparing our approach with original SAM and other medical variants like MedSAM. However, due to the submission deadline and the computational time required to train (Med)SAM from scratch, we were unable to perform these experiments. It's worth noting that our proposed training strategy is environmentally friendly, as three rounds of continual training of SAM from scratch would require a substantial amount of training time. Compared to the approximately 3 hours required per stage for a UnCLe SAM, this duration does not include the additional time needed for re-training the ViT in traditional SAM or MedSAM models.
>
> 2. **Clarification of 'UnCLe'**: We want to clarify that 'UnCLe SAM' stands for "Unleashing Continual Learning for SAM." This designation denotes our approach of leveraging continual learning techniques to unleash the performance of SAM.

---

> > ### Comment · Reviewer_TAJc · 2024-03-19
> > **Not convinced**
> >
> > 1. "the computational time required to train (Med)SAM from scratch", since SAM/MedSAM are zero-shot segmentation foundation models, which do not require additional training. Despite its limited performance on medical image segmentation tasks, it should also be added for comparison.
> > 2. I would suggest to add the "Unleashing Continual Learning for SAM" in the revised manuscript to avoid ambiguity.

---

> > > ### Author Response · Authors · 2024-03-20
> > >
> > > We appreciate your detailed observation and answer your comments in a point-by-point fashion:
> > > - In response to your feedback, we have addressed this concern by adding a figure in the Appendix (Figure 6, Appendix D), illustrating the overall problem. This figure highlights the differences between SAM, MedSAM, and our UnCLe SAM approach. SAM and MedSAM are foundation models trained in a centralized way, while UnCLe SAM is trained continuously. Meaning, that having a centralized training of MedSAM might already contain samples from our publicly used datasets, which would lead to results that are not comparable in a CL comparison, as this would be data-leakage/privacy violation in CL. To have a comparison for continuous setups, the MedSAM and UnCLe SAM should be run on a set of private datasets to ensure neither model was potentially trained on such a dataset. Unfortunately, due to limitations in accessing private datasets, we were unable to conduct comparative experiments on such datasets.
> > > - We have updated the revised manuscript and introduced the UnCLe SAM abbreviation in the text to avoid any ambiguity.

---

### Official Review · Reviewer_Mxm5 · 2024-03-04

**Confidence:** 5
**Preliminary Rating:** 4

**Summary:**

The paper introduces an adapted version of the SAM (Self-Attention Model) by modifying it to retain the feature representations of SAM's ViT (Vision Transformer) image encoder while introducing adapters based on ResNet and multi-task learning. This adaptation enhances SAM's task generalization performance in medical image detection tasks. The method shares similarities with the latest PEFT (Pseudo-Ensemble Fine-Tuning) method and additionally introduces two loss functions: BBOX regression loss and MSE category prediction loss, which can be seen as multi-task learning. These two additional task loss functions assist and optimize the true image segmentation task.

**Strengths:**

he advantages of this paper are evident: in the field of Continual Learning (CL), it leverages the latest and state-of-the-art SAM (Self-Attention Model) model and introduces a new model optimization approach based on SAM, distinct from MedSAM. The effectiveness of this method is also validated through experiments, making it worthy of attention.
This embedded Adapter and the method of introducing image BBOX detection fully utilize the prior knowledge learned by SAM in the general domain, specifically in the encoding of feature maps in ViT.

**Weaknesses:**

The drawbacks of this paper include:
1. In Figure 2, since the embeddings are derived from the feature maps encoded by ViT, I believe that these embeddings already contain rich features. Therefore, I wonder if the subsequent step of extracting features with ResNet might be redundant.
I suggest that you could use a simple attention layer (or even directly use a few fully connected layers) for fine-tuning. I truly believe that the embeddings output by ViT do not need to undergo ResNet encoding. You could conduct some ablation experiments to test whether ResNet is redundant.

2. The most crucial point is that we know SAM comprises the ViT image encoder, the Prompt encoder, and the Mask decoder for the final region output. However, this is not clearly depicted in Figure 2, which raises uncertainties. My question is whether the Segmentation Head is a standalone Mask decoder or a combination of the Mask decoder and the Prompt encoder. Additionally, what is the Prompt input for this model? If Prompt input is not necessary, SAM can omit it. I suggest that in Figure 2, you represent your improvements more clearly by basing it on the complete SAM architecture and then adding your Adapter. This way, it will be clearer to indicate your modifications rather than being confusing, leaving readers uncertain about your configuration of SAM's Prompt Encoder.

**Detailed Comments:**

Overall, the paper is well-written, and the ideas are generally acceptable. It can be published, in my opinion.

**Justification Of The Preliminary Rating:**

This paper effectively leverages the latest SAM model in Continual Learning (CL), introducing a unique optimization approach distinct from MedSAM and validated through experiments, showcasing its relevance. It utilizes prior knowledge from SAM in ViT feature map encoding for image BBOX detection. However, it lacks clarity in the need for ResNet feature extraction after ViT encoding and ambiguity in depicting SAM's components in Figure 2, raising uncertainties about the role of the Prompt encoder and Segmentation Head.

**Questions To Address In The Rebuttal:**

I do not care rebuttal. I suggest acceptance. Author should remember release their code.

---

> ### Author Response · Authors · 2024-03-14
>
> Thank you very much for your valuable suggestions. We greatly appreciate your input, and we want to inform you that we have incorporated your suggestions into the revised manuscript. Your feedback has been instrumental in improving the quality and clarity of our work. If you have any further comments or concerns, please do not hesitate to let us know.

---

### Official Review · Reviewer_LjUB · 2024-03-06

**Confidence:** 4
**Preliminary Rating:** 2
**Final Rating:** 2

**Summary:**

The paper introduced an approach uses the pre-trained SAM model for segmentation in different settings of prostate MRI dataset, in the context of continual learning. The proposed UnCLe SAM is composed by a SAM model and a ResNet-50 based segmentation header. They try to demonstrate the proposed UnCLe SAM is a robust alternative to U-Net-based approaches using multiple prostate MRI dataset. However, there are some issues need to be addressed.

**Strengths:**

The figures are well draw and the sentences are clear.
The proposed UnCLe SAM performed better than the nnUNet in the given experiment settings.
The introduction is clear for the existing works and the dataset details are well-shown.

**Weaknesses:**

1) It is not clear if this better performance was achieved by the more trainable parameters SAM model has. Similar numbers of trainable parameter settings should be given if the authors wanted to conclude the SAM-like models are better candidate of Unet-like models. Unet-like models can also have large number of trainable parameters.
2) It is not clear if the better performance was achieved by the impact of a good pretraining with enough amount of data observation. For example, if a Unet-like model with comparable capacity pretrained with similar amount of data seen by the pretrained SAM, will the discrepancies still exist on the prostate segmentation tasks?
3) If training from scratch in the continual learning setting, I highly doubt it is enough when there are only ~70 cases for Unet-like models to train on. In addition, when only using single dataset for training, the nnUNet will only take <20 cases for training, and it is hard to believe the nnUNet is trained with enough data observations in this case.  Fair comparisons between model performances should happen at least when Unet-like models are approximately trained enough.
4) Experiment setting sometimes not clear. Author should briefly state what is the ‘default setting’ of the Lifelong nnUNet framework. There are multiple settings in the paper mentioned in (Gonz´alez et al., 2023).
5) I saw the performance is the worst in DecathProst. Is this because of the resolution differences? Sinc e DecathProst has different resolutions compared with other 3 datasets, how did you do the pre-processing?

**Detailed Comments:**

Fig. 4 please use the same scale of images in order to let the reviewer do fair visual comparisons.

**Justification Of Final Rating:**

The authors partially solved my concerns, and I still think the authors' contribution was relatively limited : not be able to pre-train the baseline UNet-like models in same amount of data compared with the SAM, and thus lack of fairly comparisons. Therefore, I will keep my rating here.

**Justification Of The Preliminary Rating:**

In all, I think it is a bit not fair for the following comparisons when trying to make the proposed conclusion. Model_1: Very large number of parameters, trained on Billions of images, with an additional ResNet50 header, then trained on <20/<70 volumes of medical imaging. Model_2: less number of parameters, trained from scratch from <20/<70 medical images. I highly doubt if the baseline methods are trained enough or not.

**Questions To Address In The Rebuttal:**

Please refer to the point listed in Weaknesses section.

---

> ### Author Response · Authors · 2024-03-14
>
> Thank you for the valuable feedback. We appreciate the insights provided and have addressed each point below:
>
> 1. **Comparison of SAM-like Models with U-Net-like Models**: We acknowledge the need for clarity regarding the comparison between SAM-like models and U-Net-like models. Our intention in this study was not to assert the general superiority of SAM-like models over U-Net-like models. Instead, we aimed to introduce and evaluate a novel training strategy that capitalizes on the strengths of foundation models within a Continual Learning (CL) framework. By leveraging the knowledge encoded in the foundation models, we sought to address the challenges associated with adapting to new data distributions over time, particularly in medical imaging tasks such as prostate segmentation from MRI scans, as the title of our paper states: “UnCLe SAM: Unleashing SAM’s Potential for Continual Prostate MRI Segmentation”.
> It is important to note that our conclusion regarding the performance of SAM-like models was specific to the context of this training strategy and the metrics used for evaluation. Through comprehensive CL metrics, we demonstrated that our approach yields superior results compared to traditional nnU-Nets, particularly in the context of prostate segmentation.
> In summary, while our study showcases promising results for SAM-like models in the context of prostate segmentation from MRI scans, we emphasize the significance of our proposed training strategy within a CL framework and its potential to enhance performance in specific applications. We hope this clarification provides a comprehensive understanding of our approach and its implications within the broader context of medical image segmentation research.
>
>
> 2. **Impact of Pre-training on Better Performance**: We do not agree that pre-training with sufficient data observation may impact performance. Even with pre-training on the first prostate dataset and subsequent fine-tuning on additional datasets, as demonstrated in Table 2 and Figure 3, U-Net-based architectures, such as nnU-Nets, still struggle to achieve optimal segmentation performance. This underscores the complexity of medical image segmentation tasks and the necessity for more sophisticated approaches. We contend that the impact of pre-training with sufficient data observation may be limited in the context of medical imaging tasks. The inherently limited availability of annotated medical imaging data poses a significant challenge for pre-training based approaches.
> In contrast, SAM  benefits from pre-training on a much larger variety of data, leveraging the diverse range of medical imaging datasets to enhance its performance. Transformer architectures, such as SAM, have demonstrated favorable scaling laws when trained on large and diverse datasets. While U-Net-based architectures also exhibit scalability, we opted to leverage the already expended resources in pre-training SAM rather than introducing entirely new foundation models. This decision was driven by practical considerations, including space and time constraints in exploring the continual learning paradigm comprehensively.
> By capitalizing on SAM's pre-trained representation and integrating it within our continual learning framework, we aimed to maximize the utilization of available resources and ensure efficient adaptation to new data distributions over time. This approach not only streamlines the training process but also offers a promising avenue for addressing the challenges posed by limited annotated medical imaging data.
> In summary, while pre-training remains an essential component of many machine learning approaches, its impact on performance may be constrained by the availability of annotated data, particularly in the domain of medical imaging. Leveraging pre-trained models such as SAM presents a viable solution for maximizing the utility of existing resources and advancing the state-of-the-art in medical image segmentation.

---

> ### Author Response · Authors · 2024-03-14
>
> 3. **Training Adequacy in Continual Learning Setting**:  In addition to the segmentation performance of individual tasks, it is crucial to evaluate the sufficiency of training in the Continual Learning (CL) setting using comprehensive metrics such as FWT (Forward Transfer) and BWT (Backward Transfer). While our trained nnU-Nets achieve a Dice score of 85% on average (with 93% for ISBI, for example), which is considered decent segmentation performance for prostate tasks, it is important to note that this paper focuses primarily on evaluating performance in the CL setting.
> Forward Transfer (FWT) measures the ability of a model to adapt to new tasks or datasets while retaining the knowledge learned from previous tasks. It evaluates how well the model generalizes to unseen data distributions encountered during training, making it a critical metric for assessing the effectiveness of CL approaches. In contrast, Backward Transfer (BWT) assesses the impact of new learning on previously learned tasks. It measures the extent to which training on new tasks affects the performance of previously learned tasks, providing insights into the stability and robustness of CL models over time while considering catastrophic forgetting..
> In the context of medical imaging, where data availability is limited and new datasets may exhibit varying characteristics, FWT and BWT assume greater importance than traditional performance metrics. The ability of a model to continually adapt to new data distributions and retain previously learned knowledge is paramount for ensuring consistent and reliable performance in real-world applications.
> While individual task performance provides valuable insights into the capabilities of a model, it is the ability to adapt and learn from new data distributions over time that distinguishes CL approaches. By prioritizing FWT and BWT in our evaluation, we aim to provide a more comprehensive assessment of the effectiveness of our proposed CL framework in the context of medical image segmentation.
> In summary, while achieving high segmentation performance on individual tasks is important, evaluating performance in the CL setting using metrics such as FWT and BWT is crucial for assessing the robustness and adaptability of the models.
>
>
> 4. **Clarification of Experiment Settings**: The Lifelong nnU-Net framework's default settings involve 250 epochs with 250 steps per epoch, utilizing the optimizer and scheduler as specified in (González et al., 2023). Adhering to these standard settings ensures consistency and facilitates comparisons across experiments. We added this information in the revised manuscript. Our use of these defaults enables transparent and reproducible evaluation of our continual learning framework's performance against existing methodologies.
>
> 5. **Performance Discrepancy in DecathProst Dataset**: The lower performance observed in DecathProst can indeed be attributed to resolution differences. DecathProst has different resolutions compared to the other three datasets. Our UnCLe SAM preprocessing involves resampling to the ViT Encoder’s input size, as done in the MedSAM tutorial. Datasets are downloaded and directly used without additional preprocessing.
>
> Regarding the usage of the Lifelong nnU-Net framework, we want to assure you that the framework has been utilized correctly, as we are the developers of the Lifelong nnU-Net framework and have ensured its correct implementation. We appreciate your feedback and have taken into account the points raised to enhance the clarity and quality of our work.

---

> ### Comment · Reviewer_LjUB · 2024-03-21
>
> I still hold my doubt on the 2nd question I asked, and believe if having better pre-training with a large cohort of dataset, the UNet-like models will perform much better. The size of the datasets you mentioned used in Table 2 and Figure 3 is way less than how SAM model was trained.
>
> Also, for the 3rd question, I still believe in order to have a fair comparison, all baseline models, like the UNet-like models you mentioned, should have pretrained, or seen, enough dataset for pretraining, and then we can safely say about how we can compare using the following CL framework.
>
> I know this is hard to solve within the rebuttal time, but I will leave my comment here for your future improvement. I will keep my rating.

---

> > ### Author Response · Authors · 2024-03-22
> >
> > Thank you for your thoughtful comment regarding the potential impact of better pre-training with larger datasets on the performance of U-Net-like models. We understand your concern and appreciate your insights. Unfortunately, reliable U-Nets trained on massive datasets are not readily available in the public domain, unlike SAMs. Given the limitations of university-level compute infrastructures and the timeframe for empirical validation, training a U-Net on such massive datasets from scratch is not feasible. We acknowledge the importance of your feedback for future improvements and appreciate your understanding of the constraints we face in addressing this issue within the rebuttal period.

---

### Official Review · Reviewer_fm5E · 2024-03-08

**Confidence:** 5
**Preliminary Rating:** 2
**Recommendation:** Poster

**Summary:**

This paper proposed UnCLeSAM, an extension of the Segment Anything Model for continual learning. UnCLeSAM added to the knowledge of the pre-trained SAM continual prompting using ResNet-50 adapter. Using the image embedding output by the ViT,  the adapter generates new prompts in the form of Samples and Bounding Box to feed into the segmentation head. The approach is tested on prostate segmentation, where each task is simulated by a different dataset and compared to Lifelong nn-Unet and two CL approaches (EWC and RWalk). Results showed that UnCLeSAM outperformed comparative approaches.

**Strengths:**

-Interesting idea for CL, adding adapter that generated the prompt and freezing the ViT encoder should allow the framework to remember previous task along with adapt to new unseen tasks.
-Improved performance

**Weaknesses:**

- Experimental part is too small to support the claim, authors used only one (easy) segmentation task (segmentation of the whole prostate gland) and simulated the different tasks only by using datasets with different resolutions. Datasets are very small. Comparative methods are quite old (2017-2018)
-Experimental part is unclear

**Detailed Comments:**

- The sentence  "U-Nets encounter challenges in maintaining segmentation... clinical scenarios." Needs a reference
- It is unclear how the adapter is trained and which data is used to train it
- Why the authors did not use the all datasets used by Lifelong nnU-Net ?
- it is not clear how the continuous learning happen in the experimental part?

**Justification Of The Preliminary Rating:**

The experimental part is not clear and robust enough to justify the statement of superior performance of the method. The approach should be tested on more than one task and numerical results on each datasets should be provided.

**Questions To Address In The Rebuttal:**

Authors should
- explain how did they choose the datasets.
- provide numerical results of the CL on the single results. The results presented on table 2 for Lifelong nnU-Net are quite different from the one presented on the original paper (Lifelong nnU-Net: a framework for standardised medical continual learning)
- give more details on the experimental parts (see detailed comments)

**Special Issue:**

No

---

> ### Author Response · Authors · 2024-03-14
>
> Thank you for the detailed feedback. We appreciate the constructive comments and have addressed each point below:
>
> 1. **Dataset Selection & Usage of All Datasets**: For dataset selection and the usage of all datasets, we opted for the Prostate datasets, aligning with the experimental methodology we introduced for standardized comparison of medical continual learning research in our Lifelong nnU-Net paper. However, we made the deliberate decision not to utilize all datasets employed by Lifelong nnU-Net due to several factors. One primary consideration was the constraint posed by the limited space available in the MIDL paper (8 pages of content) restricting our capacity to include all experiments. Additionally, the significant time investment required for training all nnU-Nets, as elucidated in the Lifelong nnU-Net paper, where each continual learning (CL) method demands substantial computational resources.
> In the medical domain, evolving technologies and practices often necessitate adjustments in data acquisition methodologies. For instance, in the case of prostate segmentation, historical practices involved the use of prostate coils, which have since become obsolete. Consequently, the availability of data from these outdated methods may be limited or non-existent in contemporary datasets.
> Despite these limitations, we assure the reviewer that all experiments will be comprehensively included in the journal version of our work. This decision is motivated by our commitment to providing a thorough and exhaustive analysis of our methodology and results.
>
> 2. **Numerical Results of Continual Learning (CL)**: We have included numerical results of the CL experiments on the single results in the Appendix (see revised manuscript). Regarding the differences in results between Table 2 and the original paper on Lifelong nnU-Net, we acknowledge this variation. The discrepancy arises because we randomly split the dataset, ensuring equal distributions of train and test sets for all experiments. This differs from the random splits used in Lifelong nnU-Net, as the random splitting was performed internally with nnU-Net random splitting method. There is a significant difference in the performance between Lifelong nnU-Net and UnCLe SAM, however, the performance of Lifelong nnU-Net across different splits is similar.
>
> 3. **Reference for Sentence about U-Nets**: We have added a reference to support the statement regarding challenges encountered by U-Nets in maintaining segmentation accuracy in clinical scenarios.
>
> 4. **Training of the Adapter**: The adapter is trained using the same data as SAM. Specifically, SAM is trained solely on dataset 79, while the adapter is trained on datasets UCL, I2CVB, ISBI and DecathProst.
>
> 5. **Clarification on Continuous Learning in Experimental Part**: Our experimental setup involves training SAM and the adapter together on a dataset UCL initially. Subsequently, we freeze SAM and exclusively continue training the adapter on datasets I2CVB, ISBI, and DecathProst.

---

### Author Response · Authors · 2024-03-14

We thank all the reviewers for their insightful comments and suggestions. Their feedback has been invaluable in improving the quality of our manuscript. We greatly appreciate their time and effort in providing constructive feedback, which has helped us to refine our work for the better, and we hope our provided rebuttal solves their mentioned weaknesses. All changes have been done in a bluish color to be recognizable from the original/unmodified text.

---

### Meta-Review · Area_Chair_h4Cp · 2024-04-02

**Recommendation:** Accept (Poster)
**Confidence:** 4

**Metareview:**

All reviewers found the proposed method to be novel and the results promising. 2 weak reject, 1 weak accept, and 2 strong accept recommendations from reviewers.

---

### Decision · Program_Chairs · 2024-04-05

Accept (Poster)